# Anthelmintic Activity of Wormwood (*Artemisia absinthium* L.) and Mallow (*Malva sylvestris* L.) against *Haemonchus contortus* in Sheep

**DOI:** 10.3390/ani10020219

**Published:** 2020-01-29

**Authors:** Dominika Mravčáková, Michaela Komáromyová, Michal Babják, Michaela Urda Dolinská, Alžbeta Königová, Daniel Petrič, Klaudia Čobanová, Sylwester Ślusarczyk, Adam Cieslak, Marián Várady, Zora Váradyová

**Affiliations:** 1Institute of Animal Physiology, Centre of Biosciences of Slovak Academy of Sciences, 040 01 Košice, Slovakia; mravcakova@saske.sk (D.M.); petric@saske.sk (D.P.); boldik@saske.sk (K.Č.); 2Institute of Parasitology of Slovak Academy of Sciences, 040 01 Košice, Slovakia; komaromyova@saske.sk (M.K.); babjak@saske.sk (M.B.); dolinska@saske.sk (M.U.D.); konig@saske.sk (A.K.); 3Department of Pharmaceutical Biology with Botanical Garden of Medicinal Plants, Medical University of Wroclaw, 50-556 Wroclaw, Poland; sylwester.slusarczyk@umed.wroc.pl; 4Department of Animal Nutrition, Poznan University of Life Sciences, 60-637 Poznan, Poland; adam.cieslak@up.poznan.pl

**Keywords:** dietary treatments, plant bioactive compounds, egg counts, UHRMS, *Haemonchus contortus*

## Abstract

**Simple Summary:**

The gastrointestinal parasitic nematode *Haemonchus contortus* of small ruminants is an important target for chemoprophylaxis. Repeated use of anthelmintics in the form of synthetic drugs increases the risk of residues in food products and the development of anthelmintic resistance. However, the use of combinations of dry traditional medicinal plants as nutraceuticals is an alternative to chemotherapeutics for controlling haemonchosis in ruminants. Therefore, the aim of this study is to determine the effect of dietary supplementation with wormwood, mallow and their mix on parasitological status and inflammatory response in lambs experimentally infected with *H. contortus*. Simultaneously, the present study evaluated by the egg hatch test the in vitro anthelminthic effects of different concentrations (50–1.563 mg/mL) of the aqueous extracts of these plants. Our results revealed that the strong anthelmintic effect of both medicinal plants observed in vitro was not fully confirmed in vivo. This knowledge builds on our previously published findings and highlights that the effect of dry medicinal plants depends on the variety and synergy of plant polyphenols and the combination of bioactive compounds that together have an effect and contribute to a certain pharmacological efficacy.

**Abstract:**

The objective of this study is to evaluate the effect of dry wormwood and mallow on the gastrointestinal parasite of small ruminants *Haemonchus contortus*. Twenty-four experimentally infected lambs were randomly divided into four groups of six animals each: unsupplemented lambs, lambs supplemented with wormwood, lambs supplemented with mallow and animals supplemented with a mix of both plants. Faecal samples from the lambs were collected on day 23, 29, 36, 43, 50, 57, 64 and 75 post-infection for quantification of the number of eggs per gram (EPG). The mix of both plants contained phenolic acids (10.7 g/kg DM) and flavonoids (5.51 g/kg DM). The nematode eggs were collected and in vitro egg hatch test was performed. The aqueous extracts of both plants exhibited strong ovicidal effect on *H. contortus*, with ED50 and ED99 values of 1.40 and 3.76 mg/mL and 2.17 and 5.89 mg/mL, respectively, in the in vitro tests. Despite the great individual differences between the treated lambs in eggs reduction, the mean EPG of the untreated and treated groups did not differ (*p* > 0.05). Our results indicate that using wormwood and mallow as dietary supplements do not have a sufficient effect on lambs infected with *H. contortus*.

## 1. Introduction

The gastrointestinal nematode (GIN) infection haemonchosis is a prevalent parasitic disease associated with economic losses, lowered productivity, morbidity and mortality. *Haemonchus contortus* is highly prevalent in sheep and goats worldwide and is mainly controlled by chemoprophylaxis through the repeated application of chemotherapeutics with the risk of development of anthelmintic resistance [1].

The screening of traditional medicinal plants containing promising contents of bioactive compounds with anthelmintic activity has great potential as an alternative source of natural anthelmintics and antioxidants that may be sustainable and environmentally acceptable. Various bioactive compounds (i.e., polyphenols, flavonoids, condensed tannins) that possess an anthelmintic effect [2,3] and antibacterial and antioxidant activities have been isolated from wormwood (*Artemisia absinthium* L.) [4,5]. Many authors have reported the antioxidant and antimicrobial properties of wormwood essential oils [6,7] and the anthelmintic activity of the flavonoids quercetin and apigenin [3]. Diets containing dried wormwood as a 5%–10% replacement for rice straw also provide better quality roughage with a considerable content of crude protein [8,9,10]. The high pharmacological activity of the medicinal plant mallow (*Malva sylvestris* L.), due to the presence of amino acids, flavonoids, mucilages, terpenoids, phenol derivatives, enzymes, coumarins and sterols, is known [11,12]. Mallow has antimicrobial, antifungal and anti-inflammatory properties [13,14].

In traditional medicine, whole plants or mixtures of plants are used rather than isolated compounds, and therefore more research is needed on all types of interaction between plant constituents. The ultra-high-resolution mass spectrometry (UHRMS) analyses of dry medicinal plants or plant mixtures in our recent studies with *H. contortus* [15,16,17] identified a wide range of bioactive compounds with important pharmacological activities, mainly flavonoids, phenolic acids, diterpenes, and alkaloids. These experiments, which combined chromatographic analyses with the determination of antioxidant capacity, are helpful in identifying plants with consistent concentrations of anthelmintic and antioxidant compounds for in vitro and in vivo studies. Our previous studies showed that medicinal plant mixtures are multicomponent mixes that possess effects via a multitarget additive and synergistic mode [16,17,18].

Therefore, in the present study, we hypothesize that some medicinal plants from these mixtures are by themselves multicomponent mixes and can elicit effects via pharmacological activity on *H. contortus* infected lambs. The medicinal plants, wormwood and mallow, were chosen based on their previously described best phytotherapeutic properties and anthelmintic activity in vitro [15]. The goal is to determine the effect of dietary supplementation with wormwood and mallow on parasitological status and inflammatory parameters of lambs experimentally infected with *H. contortus*.

## 2. Material and Methods

### 2.1. Ethics Statement

All procedures and animals were cared for under European Community guidelines (EU Directive 2010/63/EU). The experimental protocol was approved by the Ethical Committee of the Institute of Parasitology of the Slovak Academy of Sciences, in accordance with national legislation in Slovakia.

### 2.2. Analysis of Bioactive Compounds

Wormwood and mallow were ground to a fine powder, and 100 mg were extracted three times in 80% MeOH for 30 min at 40 °C. The extracts were evaporated to dryness, dissolved in 2 mL of Milli-Q water (acidified with 0.2% formic acid) and purified by Solid Phase Extraction (SPE) using Oasis HLB 3cc Vac Cartrige (60 mg, Waters Corp., Milford, MA, USA). The cartridges were washed with 0.5% methanol to remove carbohydrates and then washed with 80% methanol to elute phenolics. The phenolic fraction was re-evaporated and dissolved in 1 mL of 80% methanol (acidified with 0.1% formic acid). The sample was then centrifuged (23,000× *g*, 5 min) before spectrometric analysis. All analyses were performed in triplicate for three independent samples and stored in a freezer at −20 °C before analysis. The phenolic acids and flavonoids of the plant materials were analysed by a ultra-high-resolution mass spectrometry (Dionex UltiMate 3000RS, Thermo Scientific, Darmstadt, Germany) system with a charged aerosol detector interfaced with a high-resolution quadrupole time-of-flight mass spectrometer (HR/Q-TOF/MS, Compact, Bruker Daltonik GmbH, Bremen, Germany). The metabolomes of the samples were chromatographically separated, as was described [19]. The flow rate, spectra, operating parameters, collision energy, data calibration and spectra processing were previously described [20]. The amount of the particular phenolic acids in the samples was calculated as the chlorogenic acid (CAS 327-97-9, 3-Caffeoylquinic acid) equivalent, and hyperoside (CAS 482-36-0, quercetin 3-galactoside) was used for calculating the amount of identified flavonoids. Stock solutions of hyperoside and chlorogenic acids were prepared in MeOH, as was described previously [16].

### 2.3. In Vitro Test

The in vitro egg hatch test (EHT) was performed in order to assess the ovicidal effect of aqueous extracts of wormwood and mallow and compared with the chemotherapeutic effect of thiabendazole anthelmintic drug. The nematode eggs for in vitro EHT were obtained from the untreated UNS group. The concentrations of aqueous extracts used and EHT has been previously described [15].

Chemical tests for the screening of main constituents in the medicinal plants under study were carried out in the aqueous extracts using standard procedures [21,22]. Qualitative phytochemical screening revealed the active compounds mainly tannins, flavonoids, glycosides, saponins, alkaloids, and terpenoids (Table 1).

### 2.4. Experiment In Vivo

The experiment was conducted on 24 3–4-month-old female lambs (Improved Valachian) with initial body weights of 18.67 ± 0.55 kg. The lambs were housed in common stalls on a sheep farm (Hodkovce, Slovak Republic) with free access to water. After a period of adaptation, all parasite-free lambs were infected by L3 larvae of *H. contortus* MHCo1 strain [16]. The diet of each animal consisted of oats (500 g DM/d) and meadow hay (ad libitum). Four groups of six animals based on their live-weight were established: unsupplemented lambs (UNS), lambs supplemented with stem of *A. absinthium* (ART, 1 g DM/d/lamb), lambs supplemented with flower of *M. sylvestris* (MAL, 15 g DM/d/lamb) and animals supplemented with mix of *A. absinthium* and *M. sylvestris* (ARTMAL, 16 g DM/d/lamb). The doses of plant supplements were based on the plant proportions used in our previous study [18]. The dry plants from commercial sources (AGROKARPATY, Plavnica, Slovak Republic) were mixed daily with the oats during the experimental period (75 days, D). The lambs were weighed on D0, D15, D30, D45 and D70. Faecal samples from the rectum of lambs were collected on D23, D29, D36, D43, D50, D57, D64 and D75 post-infection for quantification of the eggs per gram (EPG). The detection of strongylid eggs was performed by McMaster technique, as was previously described [23]. The blood sera samples of each animal were obtained from D15, D30, D45 and D70 [18]. Helminthological autopsy were done after 75 days of infection [16].

### 2.5. Blood Sera Analysis

Sheep immunoglobulin G (IgG), sheep immunoglobulin A (IgA) and sheep eosinophil peroxidase (EPX) were measured by ELISA kits (MyBioSource Ltd., San Diego, CA, USA). The sensitivity of the IgG, IgA and EPX kits were 0.938 ng/mL, 1.875 ng/mL and 1.0 ng/mL, respectively.

### 2.6. Statistical Analysis

Statistical analysis was performed using analysis variance (GraphPad Prism 8, GraphPad Software, Inc., San Diego, CA, USA) as repeated-measures mixed models representing the four animal groups (UNS, ART, MAL, ARTMAL) and sampling days. Differences between the animal groups were analysed by a two-way ANOVA with a Bonferroni post hoc test. Differences between the arithmetic EPG means between groups and between worm counts at dissection were analysed by Student’s *t*-tests. A logistic regression model was used to determine the ED_50_ and ED_99_ [24].

## 3. Results

### 3.1. Bioactive Compounds

The *A. absinthium* contained 6.48 g/kg DM of phenolic acids and 0.35 g/kg DM of flavonoids with greater concentrations of chlorogenic acid (3.42 g/kg DM) and 1,5-dicaffeoylquinic acid (2.12 g/kg DM) (Table 2). The *M. sylvestris* contained 0.65 g/kg DM of phenolic acids and 6.48 g/kg DM of flavonoids with higher concentrations of delphinidin-5-glucoside 3-lathyroside (1.64 g/kg DM), kaempferol-3-O-rutinoside (0.82 g/kg DM), apigenin-o-hex (1.56 g/kg DM) and coumarinic acid (0.47 g/kg DM). The mix of both plants contained 10.7 g/kg DM phenolic acids and 5.51 g/kg DM of flavonoids with greater concentrations of methyl-4-O-beta-d-glucopyranosylcaffeate (2.23 g/kg DM), 1,5-dicaffeoylquinic acid (1.64 g/kg DM), kaempferol-O-Hex (1.40 g/kg DM), apigenin-O-Hex (1.29 g/kg DM) and luteolin-O-Hex (0.70 g/kg DM).

### 3.2. In Vitro Test (EHT)

The dose-response relationships of aqueous extracts of *A. absinthium* or *M. sylvestris*, respectively, against *H. contortus* in the egg hatch test (EHT) are shown in Figure 1a,b. Both aqueous plant extracts exhibited a strong ovicidal effect on *H. contortus* in in vitro EHT. The ED_50_ and ED_99_ values were 1.40 and 3.76 mg/mL in *A. absinthium* (Figure 1a) and 2.17 and 5.89 mg/mL in *M. sylvestris* (Figure 1b), respectively. Thiabendazole at a concentration of 1.0 µg/mL has a 100% ovicidal effect.

### 3.3. Parasitological Status

The patterns of egg shedding for UNS, ART, MAL and ARTMAL are shown in Figure 2. Data from D36 were statistically compared and used to determine the reduction in egg output for ART, MAL and ARTMAL relative to UNS. Mean faecal eggs per gram (EPGs) were influenced by time from infection (*p* < 0.05), and for all groups, EPGs increased until D50 or D57, respectively. The EPGs in the lambs treated with MAL, ART and ARTMAL compared with UNS group did not differ (*p* > 0.05). The necropsy on D75 found a numerical decrease (*p* > 0.05) in the abomasal worm counts for the ARTMAL groups compared to the other groups (Figure 3).

### 3.4. Inflammatory Response

Table 3 shows the inflammatory IgG, IgA and EPX response. Serum IgG and IgA values were not influenced by treatment, time and treatment × time (*p* > 0.05). Mean serum EPX values ranged in the treated groups from 24.1 to 73.4 ng/mL, and the values were influenced by treatment and time (*p* < 0.05).

## 4. Discussion 

In the present study, phenolic compounds, flavonoids and phenolic acids among them, were detected in wormwood, mallow and a mix of both plants. Phenolic acids, including chlorogenic acid, caffeoylquinic acid derivatives, coumaric acid and methyl 4-O-beta-D-glucopyranosylcaffeate, were identified in a range from 0.65 to 10.7 g/kg DM. Chlorogenic acid and 1,5- and 4,5-dicaffeoylquinic acid possess well-known antibacterial, anthelmintic, anti-inflammatory, and antioxidant biological activities in vitro and in vivo [25,26]. Similarly, coumaric acid in *Senegalia gaumeri* leaf extract compounds has shown potential anthelmintic effects against *H. contortus* larvae [27]. The phenolic acid contents for wormwood and the mix (but not for mallow) were within the range of 3.6 to 57.3 g/kg DM, as was reported for plant mixtures used previously in infected lambs [16,17]. In relation to flavonoids with antioxidant properties, we identified mainly flavones (apigenin and luteolin), flavonols (kaempferol and quercetin) and flavanones (naringenin) [28], which may also have anthelmintic activity [3,29]. However, the total content of flavonoids in wormwood in the present study was lower (0.35 g/kg DM) versus mallow or the mix (6.48 or 5.51 g/kg DM, respectively) or compared to previous studies (9.96 and 29.5 or 41.5 g/kg DM, respectively) [16,17].

It is clear that medicinal plants that have an anthelmintic effect in vitro are often not equally effective in vivo, because there is different bioavailability, pharmacology of host animals, metabolism of bioactive compounds by rumen microflora and experimental conditions [30]. In the present experiment, the aqueous plant extracts of both medicinal plants, *A. absinthium* and *M. sylvestris*, exhibited a strong ovicidal effect on *H. contortus* in vitro, similar to the extracts of the species *Artemisia* against sheep nematodes [31,32]. However, the mean egg outputs of the UNS group compared to ART, MAL and ARTMAL groups showed no significant differences in egg reduction in lambs. Egg production by *H. contortus* females remained high (i.e., thousands EPG) until D50 post-inoculation and then decreased similarly as during the patent period of *H. contortus* in sheep [33]. The rapid reduction in egg excretion after D50 (MAL and ARTMAL) or D57 (ART), respectively, was accompanied by a lower number (not statistically) of adult *H. contortus* worms in the ARTMAL group. No significant differences in egg excretion in the treated groups may have been due to the lower content of plant biologically active compounds, especially flavonoids, compared to our previous studies [16,17]. However, relatively high SD of the means of the treated groups in the present experiment point to a potentially different treatment effect between lambs. This suggests that these plant materials could have an indirect antiparasitic effect and may promote a host’s resistance to parasitic infection in the longer term. However, the anthelmintic mechanism of action is unknown. It seems that flavonoids with antioxidant capacity, in particular, contribute to the indirect antiparasitic activity [34,35]. However, not only the content of flavonoids appears to play an important role in the anthelmintic activity of medicinal plants and their mixtures. Our results indicate that wormwood and mallow themselves are not responsible for egg and worm reduction in the lambs but probably acting in synergy with other medicinal plants and their bioactive compounds, as was done in previous mixes [17,18]. The effect of dry medicinal plants on the health of animals depends on the source of the multitarget complex bioactive compounds that work synergistically [36,37] and antagonistically [38]. An increase in the resistance of infected lambs to *H. contortus* infection was shown after the administration of mixtures of dry medicinal plants composed already of 9–13 species [16,17,18]. However, a generally great contribution to the discovery and development of new drugs is a widely applicable strategy for identifying the combinatory compounds responsible for a certain pharmacological activity of plant medicines followed by in vitro and in vivo validation [39].

Ruminal and intestinal fermentation parameters can be manipulated by supplementing a diet with medicinal plants [40]. No adverse effects of wormwood and mallow on the ruminal fermentation parameters (i.e., pH, ammonia N, methane, gas production, and volatile fatty acids) were found [18]. Mainly, pH and ammonia N can affect the release of phenolic compounds from plant materials and the growth of ruminal microbes for microbial protein synthesis [41,42]. Additionally, phenolic compounds, especially flavonoids, can improve body weight gain, growth and the quality of animal products [43]. In the present experiment, polyphenols as dietary supplements did not significantly affect the body weights or live-weight gains of the infected lambs. Dry medicinal plants in the diets of the infected lambs may [18] or may not have influenced the body weights or live-weight gains, which is consonant with a meta-analysis of gastro-intestinal nematode infection in sheep [44].

Our recent studies [16,17,18] of lambs infected with the gastrointestinal nematode *H. contortus* showed the potential value of medicinal plant mixtures to decrease egg output and worm numbers in parasitic infections of the digestive tract. However, this effect is probably not a consequence of a direct anthelmintic impact on the viability of nematodes, but an increase in the resistance of lambs to nematode infections. Additionally, a recent study also showed that a complementary vegetable mixture of plants belonging to the *Compositae*, *Cesalpinacae*, *Liliacae*, *Bromeliaceae* and *Labiatae* families used as feed at two different dosages was ineffective against gastrointestinal nematode infection [45]. It seems that mainly a combination of medicinal plants belonging to different botanical families with beneficial bioactive compounds probably contributes to slowing the dynamics of infection. In the present study, because of the low variety and synergy of plant polyphenols and the combination of bioactive compounds of wormwood and mallow, the reduction in parasitic infection intensity in the treated infected lambs was not sufficient during the 75 days of infection compared to previous studies [16,17,18]. However, it seems that mixtures of dry medicinal plants may affect the host over the longer term. Therefore, more research is needed on combinations of medicinal plants and interactions between compositions of plant mixtures for longer (90-120 days) experimental periods.

## 5. Conclusions

The data in the present study showed additional new knowledge on the anthelmintic effects of dry medicinal plants as dietary supplements. Our results indicate that using medicinal plants, even those with the best anthelmintic properties in vitro, may not have sufficient effects in vivo on *H. contortus* infected lamb.

## Figures and Tables

**Figure 1 animals-10-00219-f001:**
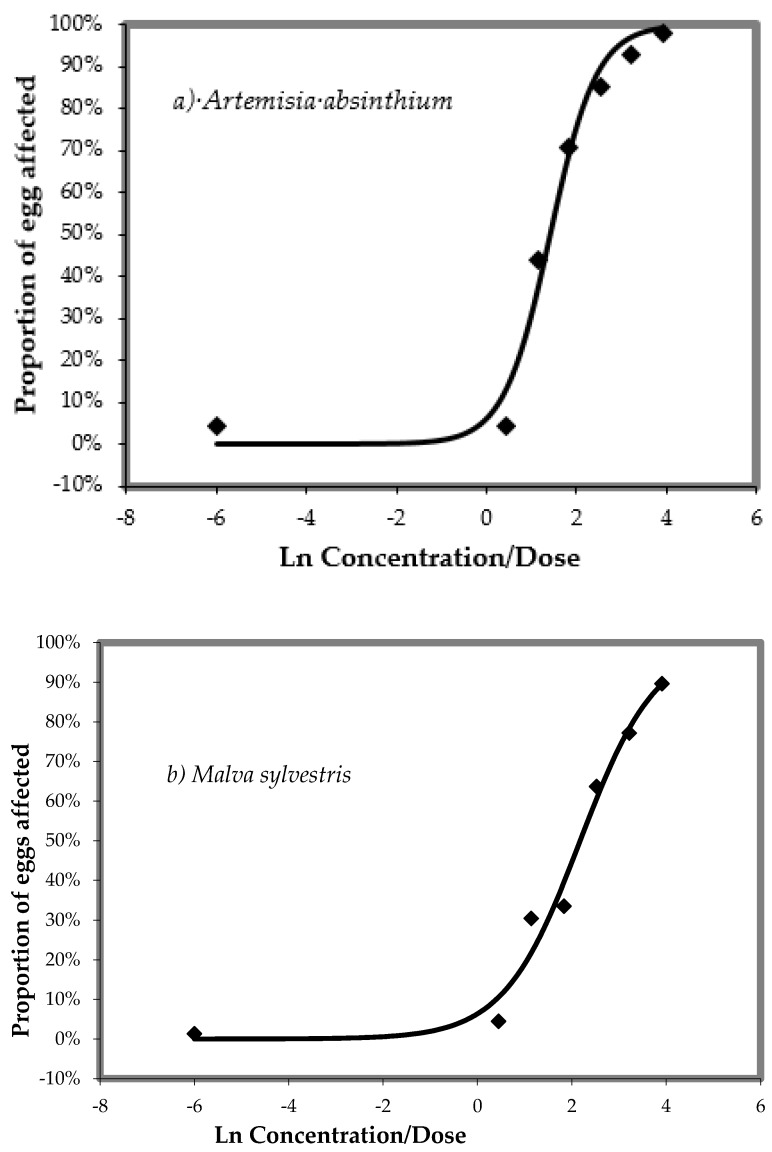
(**a**,**b**) Dose-response relationship of plant aqueous extracts against *Haemonchus contortus* in the egg hatch test (EHT) after 24 h of incubation at 26 °C.

**Figure 2 animals-10-00219-f002:**
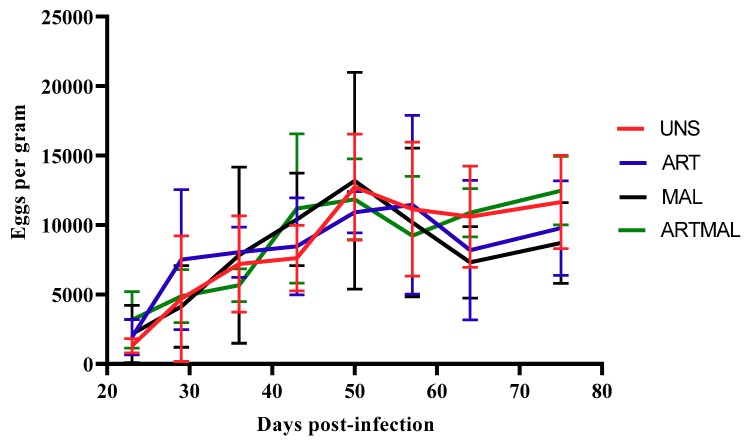
Mean faecal egg counts for the groups of lambs infected with *Haemonchus contortus* (Treatment: *p* > 0.05; time: *p* < 0.001; treatment × time: *p* > 0.05). UNS: unsupplemented; ART: *A. absinthium*; MAL: *M. sylvestris*; ARTMAL: ART plus MAL.

**Figure 3 animals-10-00219-f003:**
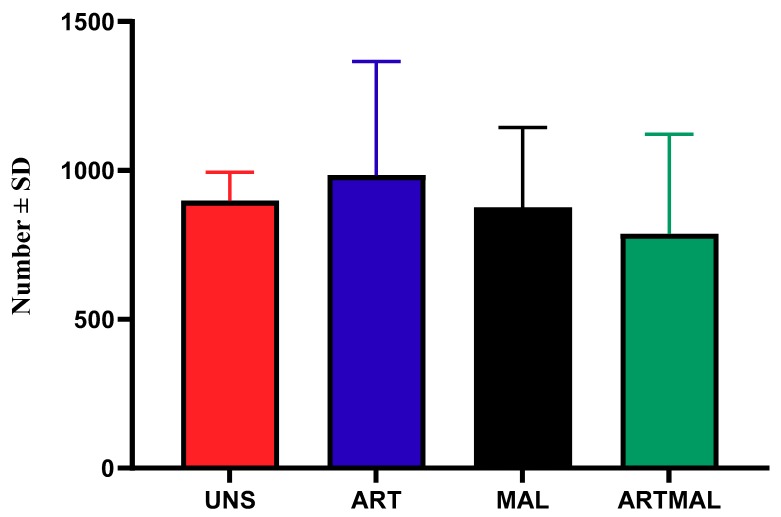
Abomasal worm counts of *Haemonchus contortus* in the lambs of each treatment at the end of the experiment (*p* > 0.05). UNS: unsupplemented; ART: *A. absinthium*; MAL: *M. sylvestris*; ARTMAL: ART plus MAL.

**Table 1 animals-10-00219-t001:** Chemical composition of the aqueous plant extracts.

Plant Species	Tannins	Flavonoids	Glycosides	Saponins	Alkaloids	Terpenoids
*A. absinthium*	+	-	-	+	-	+
*M. sylvestris*	+	+	+	-	+	+

(+): the presence of phytochemicals; (-): the absence of phytochemicals.

**Table 2 animals-10-00219-t002:** Contents of the flavonoids and phenolic acids (g/kg DM) identified in the plants and mix.

No.	RT (min)	λ_max_ (nm)	*m*/*z* [M-H]^-^	MS^2^	MS^2^ Fragments	Formula	Compound	Flavonoids	Phenolic Acids
*Artemisia absinthium*									
1	2.80		189.0759	127/0759	171/145/115	C_8_H_14_O_5_	L-(-)Malic acid diethyl ester		0.22
2	4.10	215.325	353.0877	191/0567	179/161/135	C_16_H_18_O_9_	Chlorogenic acid		3.42
3	7.80		281.1023			C_14_H_18_O_6_	ND	0.01	
4	8.00		367.1031	191/0546	173	C_17_H_20_O_9_	3-O-Feruloylquinic acid		0.08
5	8.90		279.1223	234/1009	261/217/177/199	C_15_H_20_O_5_	Artabsinolide		
6	9.10		325.1283	163	279/235	C_16_H_22_O_7_	ND		0.03
7	9.20		327.1440	279	235	C_16_H_24_O_7_	ND		
8	10.00	289.000	263.1282	201/1271	245/219/149/161/177	C_15_H_20_O_4_	Tanacetin		
9	10.20		281.1386	219/373	263/237/201	C_16_H_24_O_7_	Artabsinolide D		
10	11.00		515.1193	353/0867	191/179/135	C_25_H_24_O_12_	1,5-Dicaffeoylquinic acid		2.12
11	11.20		653.1719	345/0595	330/302	C_29_H_34_O_17_	Spinacetin 3-rutinoside	0.24	
12	11.40		477.1032	314/0415	357	C_22_H_22_O_12_	Isorhamnetin 7-glucoside	0.10	
13	11.70		515.1192	353/0869	173/179/191/155	C_25_H_24_O_12_	4,5-Dicaffeoylquinic acid		0.61
14	14.90		507.1502	413/1246	101/324/259	C_24_H_28_O_12_	Hedycoryside B		
15	15.00		511.2698	467/2775	405	C_30_H_40_O_7_	Anabsin		
16	15.50		345.1344	301/1433	257/213/187	C_19_H_22_O_6_	Diosbulbin E		
17	15.60		511.2698	245/1175	263/201	C_30_H_40_O_7_	Anabsin		
18	16.50		329.2323	211/1324	229/171/183/139	C_18_H_34_O_5_	Pinellic acid		
			Total flavonoids and phenolic acids					0.35	6.48
	*Malva sylvestris*								
1	1.60	250.320	517.1195	355/0667	193	C_21_H_26_O_15_	Ferullo-O-Hex-O-Hex		0.02
2	1.80	250.301	206.0443	144/0437		C_10_H_9_NO_4_	ND		0.17
3	7.00	523	757.1846	347/0761	329/261/509	C_32_H_39_O_21_	Delphinidin 5-glucoside 3-lathyroside	1.64	
4	7.90	308	163.0381	119/0502		C_9_H_8_O_3_	Coumaric acid		0.47
5	8.00	288	465.1046	303/0505	285/275/177	C_21_H_22_O_12_	Xeractinol	0.17	
6	8.20	520	449.1094	287/0555	259/243	C_21_H_22_O_11_	Cyanidin-O-Hex	0.28	
7	8.50	518	593.1645	431/0982	269/0460	C_27_H_31_O_15_	Pelargonidin-O-Hex-O-Hex	0.14	
8	8.70	283	687.1784	507/1142	345/0629/165		ND	0.18	
9	9.00	283	525.1246	345/0815	165/197/139	C_23_H_25_O_14_	ND	0.07	
10	9.20	287	303.0498	153/0169	125/217	C_15_H_12_O_7_	ND	0.04	
11	9.50	283	773.1781	507/1124	345/165	C_32_H_38_O_22_	ND	0.22	
12	10.00		609.1458	301/0330		C_27_H_31_O_16_	Quercetin-3-O-rutinoside	0.40	
13	10.20	268.343	447.0928	285/0386		C_21_H_20_O_11_	Kaempferol-O-Hex	0.49	
14	10.50	346	505.0981	343/0442		C_23_H_22_O_13_	Quercetin 3’-glucoside-7-acetate	0.03	
15	10.90	266.343	593.1504	285/0395		C_27_H_30_O_15_	Kaempferol-3-O-rutinoside	0.82	
16	11.10	291.346	433.1124	271/0599	151	C_21_H_22_O_10_	Naringenin-O-Hex	0.13	
17	11.25	291	287.0550	259/0596	152/201/243	C_15_H_12_O_6_	Tetrahydroxyflavone	0.30	
18	11.40	268.336	431.0978	269/0435		C_21_H_20_O_10_	Apigenin-O-Hex	1.56	
19	15.00	285.340	271.0595	151/0012	177/119	C_15_H_12_O_5_	Naringenin	0.01	
20	15.40		327.2169			C_18_H_32_O_5_	(E)-10-(8-Hydroxyoctanoyloxy)-enoic acid		
21	15.70	215.334	269.0443	151/0016	225	C_15_H_10_O_5_	Trihydroxyflavone	0.02	
			Total flavonoids and phenolic acids					6.50	0.66
Mix of *A. absinthium* and *M. sylvestris*									
1	4.00	215.325	353.0883	191/0561	173/179	C_16_H_18_O_9_	4-O-Caffeoylquinic acid		0.61
2	4.10	215.325	353.0883	191/0561	179/173	C_16_H_18_O_9_	3-O-Caffeoylquinic acid		0.74
3	5.90	215.287	355.1035	193/0498	149/134	C_16_H_20_O_9_	1-O-2’-Hydroxy-4’- methoxycinnamoyl -b-D-glucose		0.38
4	6.10	215.302	355.1038	149/0598	193/134	C_16_H_20_O_9_	1-O-Feruloylglucose		0.70
5	7.90		161.0225	133/0282		C_9_H_6_O_3_	Umbeliferone		0.40
6	8.00		323.0760	161/0221		C_15_H_16_O_8_	Mahaleboside		0.02
7	8.10	225.287	465.1033	303/177	285/0399	C_21_H_22_O_12_	Xeractinol	0.04	
8	8.30	520.000	449.1094	287/0555	259/243	C_21_H_22_O_11_	Cyanidin-O-Hex	0.03	
9	8.50		367.1025	173/0433	193/155/134	C_17_H_20_O_9_	Feruloylquinic acid		0.25
10	9.00	233.294.318	355.1034	193/0507	149/134	C_16_H_20_O_9_	Methyl-4-O-beta-D- glucopyranosylcaffeate		2.23
11	9.80	255.354	463.0882	301/0337	343	C_21_H_20_O_12_	Quercetin O-Hex	0.44	
12	9.90	252.351	609.1472	301/0331	285/0415	C_27_H_30_O_16_	Isoquercitrin O-Dhex	0.42	
13	10.30	257,4	447.0920	285/0386		C_21_H_20_O_11_	Kaempferol-O-Hex	1.40	
14	10.70	217.291.325	515.1189	353/0877	179/191	C_25_H_24_O_12_	3,5-Dicaffeoylquinic acid		0.80
15	10.80		187.0958	125/0968	169	C_9_H_16_O_4_	ND		
16	10.90	221.329	593.1520	285/0397		C_27_H_30_O_15_	Kaempferol-3-O-rutinoside	0.37	
17	11.10	217.291.325	515.1197	353/0869	191/179	C_25_H_24_O_12_	1,5-Dicaffeoylquinic acid		1.64
18	11.15	291.346	433.1124	271/0599	151	C_21_H_22_O_10_	Naringenin-O-Hex	0.19	
19	11.40	266.3	431.0976	269/0434		C_21_H_20_O_10_	Apigenin O-Hex	0.56	
20	11.50	268.343	447.0928	285/0386		C_21_H_20_O_11_	Luteolin O-Hex	0.70	
21	11.60	266.3	431.0976	269/0434		C_21_H_20_O_10_	Apigenin O-Hex	0.73	
22	11.70	215.290.325	515.119	353/0868	173/179/191	C_25_H_24_O_12_	4,5-Dicaffeoylquinic acid		0.68
23	12.40	325.0	517.1342	355/1022	353/193/149/161	C_25_H_26_O_12_	3-caffeoyl-4-dihydrocaffeoyl quinic acid		0.35
24	12.90	268.320	639.3176	519/2604	476/373/145	C_37_H_44_N_4_O_6_	Tris-trans-p-coumaroylspermine		0.50
25	13.10	218.268.339	473.1083	269/0426	413	C_23_H_22_O_11_	Apigenin -O-(Hex-Ac)	0.12	
26	13.70	325.0	517.1330	323/0759	353/193/149/161	C_25_H_26_O_12_	4-caffeoyl-3-dihydrocaffeoyl quinic acid		0.07
27	14.20	218.268.339	473.1083	269/0426	413	C_23_H_22_O_11_	Apigenin -O-(Hex-Ac)	0.22	
28	14.40	266.336	515.1187	269/0444		C_25_H_24_O_12_	Formononetin 7-O-glucoside-6’’-malonate		0.22
29	15.00	285.340	271.0595	151/0012	177/119	C_15_H_12_O_5_	Naringenin	0.07	
30	15.40		327.2169			C_18_H_32_O_5_	(E)-10-(8-Hydroxyoctanoyloxy)dec-2-enoic acid	0.03	
31	15.70	215.334	269.0443	151/0016	225	C_15_H_10_O_5_	Trihydroxyflavone	0.03	
32	17.10	222.309	785.3554	545/2397	665/502/399/145	C_46_H_50_N_4_O_8_	Tetra-trans-p-coumaroylspermine		0.47
33	19.00		373.0914	358/0681	343/329/315	C_19_H_18_O_8_	Dihydroxy—tetramethoxyflavone	0.03	
34	20.30	267.334	559.1069	269/0443	515/1172	C_26_H_24_O_14_	Apigenin 7-(2’’-acyl-6”maloylglycosyl)	0.13	0.65
			Total flavonoids and phenolic acids					5.51	10.7

No: peak numbers from UV chromatograms; RT: retention time; λ_max_: wavelengths of maximum absorption in the visible region; MS^2^: main ion; ND: not determined.

**Table 3 animals-10-00219-t003:** Inflammatory responses of the experimental groups (*n* = 6).

Item	Day	UNS	ART	MAL	ARTMAL	SD	Significance of Effects		
							Treatment (T)	Time	T × Time
IgG	15	0.627	2.15	2.31	2.23	2.54	NS	NS	NS
(ng/mL)	30	1.03	0.780	0.986	1.61	1.61			
	45	1.33	0.639	0.378	1.04	1.07			
	70	1.16	0.689	3.53	1.39	1.66			
IgA	15	0.434	1.28	0.811	0.767	0.522	NS	NS	NS
(ng/mL)	30	0.825	0.666	0.519	0.589	0.245			
	45	0.529	0.438	0.787	0.485	0.233			
	70	0.813	0.696	0.726	0.626	0.195			
EPX	15	37.5	66.6	52.2	50.0	17.9	*	*	NS
(ng/mL)	30	44.0	36.9	29.4	44.8	13.9			
	45	31.4	38.6	24.1	51.4	17.5			
	70	49.2	33.4	37.8	73.4	21.1			

UNS: unsupplemented; ART: *A. absinthium*; MAL: *M. sylvestris*; ARTMAL: ART plus MAL; EPX: eosinophil peroxidase; NS: not significant; SD: standard deviation. * *p* < 0.05.

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
