# Peer review of "Anthelmintic Activity of Wormwood (Artemisia absinthium L.) and Mallow (Malva sylvestris L.) against Haemonchus contortus in Sheep"

_animals, 2020, doi:10.3390/ani10020219_

Round 1

Reviewer 1 Report

This is an interesting topic as the parasitic worms of sheep, horses and livestock have developed resistance to anthelmintic drugs, which is a growing threat to the health of animals. Like antibiotics, anthelmintic drugs must be used restrictively to stay active for as long as possible. It is crucial for the future to find long-term sustainable practices for parasite control, including alternative methods such plants, be applied to slow down the development of resistance without at the same time leading to increased parasite damage. Over all the manuscript is well written, but some part are lacking structure, se comments below.

An overall comments:

The authors needs to discuss why only EHT used for the in vitro assay. To be able to investigate the effect of anthelmintic compound in vitro should EHT be combined with LDA. For instance will substances belonging to the Macrocyclic Lactones have no effect on an EHT due the mechanism of action of the anthelmintic effect. Please explain why LDA was not used.

The authors discuss synergetic effects of plat extracts but they do not mention any anthelmintic mechanism of action. Include a section in the discussion if the mechanism of these extracts are known.

Simply summary lack structure. The aim, study design and results are not clear. It needs to be rewritten.

Line 39 Need to state how the in vitro test was performed The aqueous extracts of both plants exhibited strong ovicidal effect on H. contortus, with ED50 and ED99 values of 1.40 and 3.76 mg/mL and 2.17 and 5.89 mg/mL, respectively, in the in vitro tests.

Line 41. Please clarify this sentence. Differences between which groups? Despite the great individual differences between the treated lambs in eggs reduction, the EPG of the infected groups did not differ (p > 0.05).

Line 80 add in vitro The medicinal plants wormwood and mallow were chosen based on their previously described best phytotherapeutic properties and anthelmintic activity [15].

Line 92 this paragraph need rephrasing. The structure is not logic, ie lambs need to be infected before collection of eggs. Also state where the MHco3 strain is from. Are eggs collected from the same lambs that are included in the trail?

Line 103 2.3. Animals, Diets and Experimental Design Same comment as above. If the same labs were used for collection of eggs for the EHT should this paragraph be before the EHT.

Line 115 (ARTMAL, 16 g DM/d/lamb. The end bracket is missing.

Line 167 Write ELISA with capital letters

Line 184 The materials and methods and the results section are not following the same structure. Eg results section starts with bioactive compounds and the materials and method with EHT.

Line 203 rephrase and include influenced by time from infection….  Mean faecal eggs per gram (EPGs) were influenced by time (p 203 < 0.05), and for all groups EPGs increased until D50 or D57, respectively.

Line 204. Rephrase as this is not significant result do describe the differences observed as trends. The EPGs in the lambs 204 treated with MAL and ART decreased from D50 and/or D57 up to D64 and then slightly increased 205 until the end of the experiment. The EPGs in the lambs treated with ARTMAL decreased from D50 206 until D57 and then slightly increased until the end of the experiment. Eggs per gram for UNS from 207 D50 to D75 remained almost stable.

Line 208 unclear. Was there a difference between the treated groups and the controls? Which groups are compared? Great individual differences in eggs reduction were observed 208 between the treated lambs, but the EPG of the groups did not differ (p > 0.05).

Line 216 all treatments are confusing. Do you mean all experimental groups? Please screen the manuscript for this typo.

Line 248 this sentence need rephrasing. However, the egg outputs of the UNS, ART, MAL and ARTMAL groups showed no significant differences in eggs reduction in lambs.

Line 249 define “high” Egg production by H. contortus females remained high until D50 post-inoculation and then decreased similarly as during the patent period of H. contortus in sheep

Line 253 consider rephrasing the message of this sentence is unclear The slight reduction in egg excretion in the treated groups may have been due to the lower content of plant biologically active compounds, especially flavonoids, compared to our previous studies [16,17].

Line 255 This statement is vague as the McMaster technique is self has high variability. This needs to be rephrase and also include the “normal” large SD observed when calculating EPG. However, larger SD of the treated groups in the present experiment showed great differences between lambs in eggs reduction. This suggests that these plant materials could have an indirect antiparasitic effect and may promote a host’s resistance to parasitic infection in the longer term.

Line 258

This statement is not evaluated in this study. Consider rephrasing. This indirect antiparasitic effect, which was present in some individual lambs during the experiment, probably depends on the development of protective acquired immune responses that depend on age, nutritional status and host genotype [32].

showed additional new knowledge on the anthelmintic effects of dry medicinal plants as dietary supplements.

Line 309 rephrase. Do you mean in reducing EPG? Our results indicate that using only one or two medicinal plants, even those with the best anthelmintic properties in vitro, did not have a sufficient effect via their pharmacological activity on H. contortus infected lambs.

Line 312 Is this a conclusion from this paper? This knowledge builds on our previously published findings and highlights the fact that the effect of medicinal plants depends on the variety and synergy of plant polyphenols and the combination of bioactive compounds that taken together contribute to a certain pharmacological efficacy.

Please specify what is promising The results of this study points to the promising beneficial effects of using dry medicinal plant mixtures against gastrointestinal nematodes of sheep, but more research is needed on their combinations and interactions.

Is it possible to make one figure of 1 and 2?

Author Response

Revision Note

Editor Comments: 1. Please use the version of your manuscript found at the above link for your revisions (also attached), as the editorial office have made formatting changes to your original submission.

AUTHORS: We used the attached version from editorial office for our revision.  

Editor Comments: 2. Please appropriately revise the Section 2-Material and methods according to attached check report.

AUTHORS: The Section 2 Material and methods has been revised according to attached check report from editorial office.

Editor Comments: 3. Please insert all figures and tables into the main text close to their first citation.

AUTHORS: All figures and tables have been inserted close to their first citation according to comments of Editor.

REVIEWER 1 (R1):

R1: Overall comments:

The authors need to discuss why only EHT used for the in vitro assay. To be able to investigate the effect of anthelmintic compound in vitro should EHT be combined with LDA. For instance will substances belonging to the Macrocyclic Lactones have no effect on an EHT due the mechanism of action of the anthelmintic effect. Please explain why LDA was not used.

AUTHORS: In vitro results of both plants have been already published (Váradyová et al. 2018). This study includes the results of both in vitro tests. However, in vitro EHT results are expressed as a single concentration effect (without ED50), whereas in vitro LDT data have been reported as full dose responses with LD50. For this reason, we decided to repeat only EHT (with full dose response and ED50) for both plants.

R1: The authors discuss synergetic effects of plant extracts but they do not mention any anthelmintic mechanism of action. Include a section in the discussion if the mechanisms of these extracts are known.

AUTHORS: Lines 243-249:The anthelmintic mechanism of action is unknown. From our studies to date, we only assume that the administration of medicinal plants has an indirect effect on parasites (by boosting host immunity) rather than a direct anthelmintic effect. Sentence: "However, the anthelmintic mechanism of action is unknown." has been added to Discussion.

R1: Simply summary lack structure. The aim, study design and results are not clear. It needs to be rewritten.

AUTHORS: Lines 20-24: The sentences: “Therefore, the aim of this study was determine the effect of dietary supplementation with wormwood, mallow and their mix on parasitological status, hemoglobin, albumin and inflammatory response in lambs experimentally infected with H. contortus. Simultaneously the present study has evaluated the effect the aqueous extracts for in vitro egg hatch test in a wide range of concentrations (50 -1.563 mg/mL).” have been added to the Abstract.

R1: Line 39 Need to state how the in vitro test was performed The aqueous extracts of both plants exhibited strong ovicidal effect on H. contortus, with ED50 and ED99 values of 1.40 and 3.76 mg/mL and 2.17 and 5.89 mg/mL, respectively, in the in vitro tests.

AUTHORS: Lines 35-36: The sentence: “The nematode eggs were collected and in vitro egg hatch test was perfomed.” has been added to the Abstract.

R1: Line 41. Please clarify this sentence. Differences between which groups? Despite the great individual differences between the treated lambs in eggs reduction, the EPG of the infected groups did not differ (p > 0.05).

AUTHORS: Lines 38-40: The sentence was changed as follows:” Despite the great individual differences between the treated lambs in eggs reduction, the mean EPG of the untreated and treated groups did not differ (p > 0.05).”

R1: Line 80 add in vitro The medicinal plants wormwood and mallow were chosen based on their previously described best phytotherapeutic properties and anthelmintic activity [15].

AUTHORS: Lines 77-78: The “ in vitro” was added into the text.

R1: Line 92 this paragraph need rephrasing. The structure is not logic, ie lambs need to be infected before collection of eggs. Also state where the MHco3 strain is from. Are eggs collected from the same lambs that are included in the trail?

AUTHORS: Line 104:The structure of the Materials and Methods chapter (Chapters 2.3 and 2.4) has been reorganized. Sentence: "The nematode eggs for in vitro EHT were obtained from the untreated UNS group." has been added to Chapter 2.4. A reference literature to the MHCo1 strain (the strain which has been used for infection of experimental animals) was added to Chapter 2.3.

R1: Line 103 2.3. Animals, Diets and Experimental Design Same comment as above. If the same labs were used for collection of eggs for the EHT should this paragraph be before the EHT.

AUTHORS: The structure of the Materials and Methods chapter (Chapters 2.3 and 2.4) has been reorganized according to the reviewer’s comment.

R1: Line 115 (ARTMAL, 16 g DM/d/lamb. The end bracket is missing.

AUTHORS: Line 109: The bracket was added (ARTMAL, 16 g DM/d/lamb).

R1: Line 167 Write ELISA with capital letters

AUTHORS: This sentence was removed from manuscript (check report from editorial office)

R1: Line 184 The materials and methods and the results section are not following the same structure. Eg results section starts with bioactive compounds and the materials and method with EHT.

AUTHORS: We revised the structure of Materials and Methods to be in accordance with Results. In revised manuscript both sections (i.e., Materials and Methods and Results) start with bioactive compounds.

R1: Line 203 rephrase and include influenced by time from infection….  Mean faecal eggs per gram (EPGs) were influenced by time (p < 0.05), and for all groups EPGs increased until D50 or D57, respectively.

AUTHORS: Lines 163-164: The sentence was changed as follows:” Mean faecal eggs per gram (EPGs) were influenced by time from infection (p < 0.05), and for all groups EPGs increased until D50 or D57, respectively.

R1: Line 204. Rephrase as this is not significant result do describe the differences observed as trends. The EPGs in the lambs 204 treated with MAL and ART decreased from D50 and/or D57 up to D64 and then slightly increased 205 until the end of the experiment. The EPGs in the lambs treated with ARTMAL decreased from D50 206 until D57 and then slightly increased until the end of the experiment. Eggs per gram for UNS from 207 D50 to D75 remained almost stable.

AUTHORS: Lines 164-166: The sentences have been rephrased as follows: “The EPGs in the lambs treated with MAL, ART and ARTMAL compared with UNS group did not differ (p > 0.05).”

R1: Line 208 unclear. Was there a difference between the treated groups and the controls? Which groups are compared? Great individual differences in eggs reduction were observed 208 between the treated lambs, but the EPG of the groups did not differ (p > 0.05).

AUTHORS: The sentence was unclear sentence was removed from the text.

R1: Line 216 all treatments are confusing. Do you mean all experimental groups? Please screen the manuscript for this typo.

AUTHORS: Line 203: “all treatments” was changed on “all experimental groups”

R1: Line 248 this sentence need rephrasing. However, the egg outputs of the UNS, ART, MAL and ARTMAL groups showed no significant differences in eggs reduction in lambs.

AUTHORS: Line 235-236:The sentence was changed as follows: ” However, the mean egg outputs of the UNS group compared to ART, MAL and ARTMAL groups showed no significant differences in eggs reduction in lambs.”

R1: Line 249 define “high” Egg production by H. contortus females remained high until D50 post-inoculation and then decreased similarly as during the patent period of H. contortus in sheep

AUTHORS: Lines 236-238: The sentence was changed as follows:” Egg production by H. contortus females remained high (i.e., thousands EPG) until D50 post-inoculation and then decreased similarly as during the patent period of H. contortus in sheep [33].

R1: Line 253 consider rephrasing the message of this sentence is unclear The slight reduction in egg excretion in the treated groups may have been due to the lower content of plant biologically active compounds, especially flavonoids, compared to our previous studies [16,17].

AUTHORS: Lines 241-243: The sentence was rephrased as follows: ”No significant differences in egg excretion in the treated groups may have been due to the lower content of plant biologically active compounds, especially flavonoids, compared to our previous studies [16,17].”

R1: Line 255 This statement is vague as the McMaster technique is self has high variability. This needs to be rephrase and also include the “normal” large SD observed when calculating EPG. However, larger SD of the treated groups in the present experiment showed great differences between lambs in eggs reduction. This suggests that these plant materials could have an indirect antiparasitic effect and may promote a host’s resistance to parasitic infection in the longer term.

AUTHORS: Lines 243-246: The sentences were rephrased as follows: “However, relatively high SD of the means of the treated groups in the present experiment points to a potentially different treatment effect between lambs. This suggests that these plant materials could have an indirect antiparasitic effect and may promote a host’s resistance to parasitic infection in the longer term.”

R1: Line 258 This statement is not evaluated in this study. Consider rephrasing. This indirect antiparasitic effect, which was present in some individual lambs during the experiment, probably depends on the development of protective acquired immune responses that depend on age, nutritional status and host genotype [32].

showed additional new knowledge on the anthelmintic effects of dry medicinal plants as dietary supplements.

AUTHORS: The sentence has been removed from the manuscript.

R1: Line 309 rephrase. Do you mean in reducing EPG? Our results indicate that using only one or two medicinal plants, even those with the best anthelmintic properties in vitro, did not have a sufficient effect via their pharmacological activity on H. contortus infected lambs.

AUTHORS: Lines 297-299: The sentence was rephrased as follows: ” Our results indicate that using only one or two medicinal plants, even those with the best anthelmintic properties in vitro, did not have a sufficient egg/adult reduction effect via their pharmacological activity on H. contortus infected lambs.”

R1: Line 312 Is this a conclusion from this paper? This knowledge builds on our previously published findings and highlights the fact that the effect of medicinal plants depends on the variety and synergy of plant polyphenols and the combination of bioactive compounds that taken together contribute to a certain pharmacological efficacy.

Please specify what is promising The results of this study points to the promising beneficial effects of using dry medicinal plant mixtures against gastrointestinal nematodes of sheep, but more research is needed on their combinations and interactions.

AUTHORS: The conclusion of this study is directly supported by our previous results obtained from experiments with mixtures of medicinal plants used as herbal nutraceuticals. (see reference 16,17 and 18). The last sentence from the Conclusion has been removed.

R1: Is it possible to make one figure of 1 and 2?

AUTHORS: Figure 1 and 2 were merged into Figure 1a-b on page 8.

We really appreciate your valuable comments and your effort with revising of our manuscript. Thank you very much and we hope our corrections and additions have improved the manuscript sufficiently for publication.

Reviewer 2 Report

General comment:

This interesting manuscript evaluated the anthelmintic activity in vitro and in vivo and the effects on the health and growth of treated animals of two medicinal plants. The study is very interesting, nonetheless, it needs some changes, additions, and corrections before it can be accepted. The methods are often not completely reported and, in some cases, they have not been sufficiently detailed. This may not allow the repeatability of this study. The discussion and conclusions are also incomplete, and they are probably affected by the neglect that the authors use for some aspects which instead could be fundamental to clarify the different results observed in vitro and in vivo.

In many parts of the manuscript, the English language needs to be revised.

More detailed comments are reported below.

Title:

The title is not sound and should be changed (e.g. Anthelmintic Activity of Wormwood (Artemisia absinthium L.) and Mallow (Malva sylvestris L.) against Haemonchus contortus in Sheep.

Simple Summary:

Line 23: Please replace “in the both plants” with “in both plants”.

Lines 25-30: These two sentences are just repetition.

Introduction

Line 49: Please correct the English of this sentence.

Line 60: Thujone is not an essential oil but an active principle that is highly toxic to mammals. Please, correct.

Material and methods

Lines 100-102: authors should detail the origin of the plants (commercial products? collected?). If collected, authors should detail where (area, location, altitude, soil type, etc., etc.), when (seasonal period), and how plants were collected and identified, the method used for drying them, where plant voucher specimens were placed, and the exact procedure they used for obtaining the aqueous extracts.

Lines 119-120: The method used for the detection of strongylid eggs should be detailed.

Lines 125-162: Chemical analysis of plant extracts was not performed, while it is fundamental to know if the extract used in vitro in this study differed greatly in their compositions compared to the plants of origin. In fact, this can help to know if the different efficacy observed in vitro and in vitro in this study should be attributed to a different composition of the plants and the respective extracts or to some other reasons.

Discussion and conclusions

The topic could be much more complex than what emerges from the discussion and conclusions of this study and more complex studies are probably necessary before excluding the anthelminthic properties of these plants in vivo, precluding their potential use for this purpose in the future. In fact, along with the content and chemical characterization of the chemical classes or pure compounds endowed with potential anthelminthic activity, it would also be necessary to evaluate in these same plants the simultaneous presence of other substances with antagonistic effects. Moreover, the extracts used in vitro in this study could differ greatly in their composition compared to the plants of origin. Furthermore, the complexity of the physiology of the gastrointestinal system of ruminants makes it particularly difficult to evaluate the potential efficacy of the same plant when administered fresh after the harvest or, as done by the authors in this study, after its drying (whose methods, moreover, have not been shown in this study, confirming once more that the authors probably consider unimportant some aspects that, on the contrary, could have a great relevance).

Tables and Figures

The number of tables and, above all, of figures is excessive and should be reduced, for example by combining several figures together or by presenting part of these data as supplementary materials.

Author Response

Revision Note

Editor Comments: 1. Please use the version of your manuscript found at the above link for your revisions (also attached), as the editorial office have made formatting changes to your original submission.

AUTHORS: We used the attached version from editorial office for our revision.  

Editor Comments: 2. Please appropriately revise the Section 2-Material and methods according to attached check report.

AUTHORS: The Section 2 Material and methods has been revised according to attached check report from editorial office.

Editor Comments: 3. Please insert all figures and tables into the main text close to their first citation.

AUTHORS: All figures and tables have been inserted close to their first citation according to comments of Editor.

REVIEWER 2 (R2):

R2: Title: The title is not sound and should be changed (e.g. Anthelmintic Activity of Wormwood (Artemisia absinthium L.) and Mallow (Malva sylvestris L.) against Haemonchus contortus in Sheep.

AUTHORS: The title of manuscript has been changed according to R2 suggestion on: “Anthelmintic Activity of Wormwood (Artemisia absinthium L.) and Mallow (Malva sylvestris L.) against Haemonchus contortus in Sheep.”

R2: Simple Summary:

Line 23: Please replace “in the both plants” with “in both plants”.

Lines 25-30: These two sentences are just repetition.

AUTHORS: Lines 20-24: Simple Summary was changed and this sentence was removed from manuscript.

Line 25-30: The sentence was removed to be avoiding repetition.

R2: Introduction:

Line 49: Please correct the English of this sentence.

Line 60: Thujone is not an essential oil but an active principle that is highly toxic to mammals. Please, correct.

AUTHORS: Line 46-47: The sentence was changed on: “The gastrointestinal nematode (GIN) infection haemonchosis is a prevalent parasitic disease associated with economic losses, lowered productivity, morbidity and mortality”

Lines 56-57: The sentence was corrected: “Many authors have reported the antioxidant and antimicrobial properties of the wormwood essential oils…”

R2: Material and methods:

Lines 100-102: authors should detail the origin of the plants (commercial products? collected?). If collected, authors should detail where (area, location, altitude, soil type, etc., etc.), when (seasonal period), and how plants were collected and identified, the method used for drying them, where plant voucher specimens were placed, and the exact procedure they used for obtaining the aqueous extracts.

AUTHORS: The dry medicinal plants were from commercial sources (AGROKARPATY, Plavnica, Slovak Republic) as was stated in the manuscript on Line 110. Preparation of aqueous plant extracts was as described previously Váradyová et al., 2018 (Ovicidal and larvicidal activity of extracts from medicinal-plants against Haemonchus contortus. Exp. Parasitol. 2018, 195, 71–77). A detailed description could not be provided due to acceptance of a check report from editorial office.

R2: Lines 119-120: The method used for the detection of strongylid eggs should be detailed.

AUTHORS: Lines 114-115: The sentence has been change as follows: “The detection of strongylid eggs was performed by McMaster technique as was previously described [21].” A detailed description could not be provided due to acceptance of a check report from editorial office.

R2: Lines 125-162: Chemical analysis of plant extracts was not performed, while it is fundamental to know if the extract used in vitro in this study differed greatly in their compositions compared to the plants of origin. In fact, this can help to know if the different efficacy observed in vitro and in vitro in this study should be attributed to a different composition of the plants and the respective extracts or to some other reasons.

AUTHORS: The authors agree with the idea of the opponent, however chemical test for the screening and identification of bioactive chemical constituents in both medicinal plants were carried out previously (Váradyová et al. 2017 - see reference 18) from dry materials using standard procedures.

R2: Discussion and conclusions

The topic could be much more complex than what emerges from the discussion and conclusions of this study and more complex studies are probably necessary before excluding the anthelminthic properties of these plants in vivo, precluding their potential use for this purpose in the future. In fact, along with the content and chemical characterization of the chemical classes or pure compounds endowed with potential anthelminthic activity, it would also be necessary to evaluate in these same plants the simultaneous presence of other substances with antagonistic effects. Moreover, the extracts used in vitro in this study could differ greatly in their composition compared to the plants of origin. Furthermore, the complexity of the physiology of the gastrointestinal system of ruminants makes it particularly difficult to evaluate the potential efficacy of the same plant when administered fresh after the harvest or, as done by the authors in this study, after its drying (whose methods, moreover, have not been shown in this study, confirming once more that the authors probably consider unimportant some aspects that, on the contrary, could have a great relevance).

AUTHORS: The authors agree with the reviewer conclusion, that this topic is much more complex. Our current and previous results, which are only part of the mosaic, confirm this. The path to alternative options to the currently used anthelmintics is long and may last for more than a decade(s) under the currently limited conditions for science and research. Therefore, any new experience, whether positive or negative, can bring hope to the closed ring of parasite-chemical treatment-resistance.

R2: Tables and Figures

The number of tables and, above all, of figures is excessive and should be reduced, for example by combining several figures together or by presenting part of these data as supplementary materials.

AUTHORS: Figure 1 and 2 were merged into Figure 1a-b on page 8.

AUTHORS: The English has been revised throughout the whole manuscript by a native English language editor.

We really appreciate your valuable comments and your effort with revising of our manuscript. Thank you very much and we hope our corrections and additions have improved the manuscript sufficiently for publication.

Round 2

Reviewer 2 Report

General Comment:

Although some important changes were made by the authors in the revision of this manuscript, there are still numerous shortcomings, and statements and methods that are not properly correct or scientifically correct. Furthermore, the number of tables and figures continues to be too high and not justified by the data presented in this study. Therefore, Table 1 and Figures 4 and 5 should be deleted. Tables 2, 3 and 4 should be combined into a single Table. The section on the in vitro anthelmintic assessment (Egg Hatch Test) has no meaning without showing the chemical composition of the extracts used. Therefore, as already requested in the first review, it is mandatory that the authors provide the composition (or at least the main constituents) of the aqueous extracts used or, alternatively, they should necessarily delete this section from the study and change the results and discussion accordingly.

The conclusions should be completely revised because it is only made by not scientifically sound assumptions.

Moreover, chemical characterization is also not properly performed and correctly interpreted. In fact, authors quantified the total phenolic acids and total flavonoids expressing them as two compounds chosen as representative of these two classes (why them and not others?), without however precisely identifying the compounds actually present in the plants.

Or, better, they provide a list of identified compounds, but from mass spectra analysis alone, although in high resolution (the method used in this study), it is not possible establishing the exact isomers.

In fact, the mass provides only the molecular formula of the compound, to which, however, several isomeric substances my correspond.

For example: in Table 2 the compound 11 is identified as spinacetin 3 rutinoside, but at best authors could affirm that it is spinacetin (in fact, assuming that the aglycone (with its 2 methoxyls) has been correctly identified, how authors can affirm that the sugar is bound precisely in 3? How authors can affirm that it is rutinose (glucose that carries a rhamnose in 6). This glucose could also be e.g. galactose and rhamnose could be e.g. fucose, and the mass spectrum would be the same).

Furthermore, among identified compounds, several anthocyanins are included. However, in a non-acidic environment as used by the authors (extraction with water and methanol), these compounds are not very stable and tend to degrade. For this reason, the quantification (even excluding the above limits) may have been underestimated due to the loss of part of these compounds.

Other suggested changes and comments are reported below:

Line 20: change with: Therefore, the aim of this study was to determine the effect of dietary supplementation with…

Line 23: change with: the present study evaluated by the egg hatch test the in vitro anthelminthic effects of different concentrations (50 -1.563 mg/mL) of the aqueous extracts of these plants

Line 54: essential oils are extracts and not chemical compounds

Line 88: no extraction procedure was found in the reported reference.

Lines 252-254: please add “and antagonistically” and insert a relative reference at the end of this sentence.

Line 270: what is the meaning of the sentence “Figure 4 shows that one or two medicinal plants did not have a beneficial effect on the haematological parameters” one plant, both plants the mix of the two plants, what???

Lines 297-299: please replace this sentence with “Our results indicate that using medicinal plants, even those with the best anthelmintic properties in vitro, may not have a sufficient effects in vivo on H. contortus infected lamb.

Lines 299-302: this last sentence should be deleted

Author Response

Revision Note

REVIEWER 2 (R2):

R2: The number of tables and figures continues to be too high and not justified by the data presented in this study. Therefore, Table 1 and Figures 4 and 5 should be deleted. Tables 2, 3 and 4 should be combined into a single Table.

AUTHORS: Tables 1 and Figures 4 and 5 were deleted according to R2 comments.

Tables 2, 3, and 4 have been combined into a single Table 2 (Pages 5-7) according to R2 comments.

R2: The section on the in vitro anthelmintic assessment (Egg Hatch Test) has no meaning without showing the chemical composition of the extracts used. Therefore, as already requested in the first review, it is mandatory that the authors provide the composition (or at least the main constituents) of the aqueous extracts used.

AUTHORS: We agree with Reviewer comment that the composition (or at least the main constituents) of the aqueous extracts used is mandatory. Therefore:

Lines 112-115: We added “Chemical tests for the screening of main constituents in the medicinal plants under study were carried out in the aqueous extracts using standard procedures [21,22]. Qualitative phytochemical screening revealed the active compounds mainly tannins, flavonoids, glycosides, saponins, alkaloids and terpenoids (Table 1).”

Line 116: We added Table 1: “Chemical composition of the aqueous plant extracts”

The references [21,22] were included into the List of References:

Yadav, R.N.S.; Agarwala, M. Phytochemical analysis of some medicinal plants. J. Phytol. 2011, 3, 10–14. Jaradat, N.; Hussen, F.; Al Ali, A. Preliminary Phytochemical Screening, Quantitative Estimation of Total Flavonoids, Total Phenols and Antioxidant Activity of Ephedra alata Decne. J. Mater. Environ. Sci. 2015, 6(6), 1771–1778.

R2: The conclusions should be completely revised because it is only made by not scientifically sound assumptions.

AUTHORS: Lines 268-272:

The Conclusion was completely revised according to R2 comments.

R2: Authors quantified the total phenolic acids and total flavonoids expressing them as two compounds chosen as representative of these two classes (why them and not others?), without however precisely identifying the compounds actually present in the plants.

AUTHORS: We agree with Reviewer 2 comment that a list of the components from quantitative analysis of plant secondary metabolites would be useful. However, the main aim of this study was to determine the in vitro and in vivo effects of two plants (or mixture) on parasitic infection. A complete quantitative biochemical analysis of bioactive components would not alter the parasitological results of this study. Our data were only intended to indicate that well-known bioactive components with anthelmintic and antioxidant parameters were present in both medicinal plants used in this study.

R2: Line 20: change with: Therefore, the aim of this study was to determine the effect of dietary supplementation with…

AUTHORS: Line 20: The sentence was changed according to R2 comments.

R2: Lines 23: change with: the present study evaluated by the egg hatch test the in vitro anthelminthic effects of different concentrations (50 -1.563 mg/mL) of the aqueous extracts of these plants.

AUTHORS: Lines 22-24: The sentence was changed according to R2 comments.

R2: Line 54: essential oils are extracts and not chemical compounds

AUTHORS: Line 54: “essential oils” has been removed from parenthesis of the text according to R2 comments.

R2: Line 88: no extraction procedure was found in the reported reference.

AUTHORS: Line 88: Confused Reference was deleted.

Lines 88-96: Extraction procedure has been involved into the text: “Wormwood (Artemisia absinthium L.) and mallow (Malva sylvestris L.) were ground to a fine powder, and 100 mg were extracted three times in 80% MeOH for 30 min at 40 °C. The extracts were evaporated to dryness, dissolved in 2 mL of Milli-Q water (acidified with 0.2% formic acid) and purified by Solid Phase Extraction (SPE) using Oasis HLB 3cc Vac Cartrige (60 mg, Waters Corp., Milford, MA). The cartridges were washed with 0.5% methanol to remove carbohydrates, and then washed with 80% methanol to elute phenolics. The phenolic fraction was re-evaporated and dissolved in 1 mL of 80% methanol (acidified with 0.1% formic acid). The sample was than centrifuged (23 000 × g, 5 min) before spectrometric analysis. All analyses were performed in triplicate for three independent samples and stored in a freezer at -20 °C before analysis.”

R2: Lines 252-254: please add “and antagonistically” and insert a relative reference at the end of this sentence.

AUTHORS: Lines 234-236: “and antagonistically [38].” has been added at the end of the sentence according to R2 comments. Reference has been included into the List of References:

Xutian, S.; Zhang, J.; Louise, W. “New exploration and understanding of traditional Chinese medicine,” Am. J. Chinese Med. 2009, 37, 411–426.

R2: Line 270: what is the meaning of the sentence “Figure 4 shows that one or two medicinal plants did not have a beneficial effect on the haematological parameters” one plant, both plants the mix of the two plants, what???

AUTHORS: Confused sentences (concerning Figures 4, 5 and relative References) have been removed from the manuscript.

R2: Lines 297-299: please replace this sentence with “Our results indicate that using medicinal plants, even those with the best anthelmintic properties in vitro, may not have a sufficient effects in vivo on H. contortus infected lamb.

AUTHORS: Lines 270-272: The sentence was replaced according to R2 comments.

R2: Lines 299-302: this last sentence should be deleted

AUTHORS: The last sentence has been deleted according to R2 comments.
